# Over Expression of the Cyanobacterial Pgr5-Homologue Leads to Pseudoreversion in a Gene Coding for a Putative Esterase in Synechocystis 6803

**DOI:** 10.3390/life10090174

**Published:** 2020-09-03

**Authors:** Ketty Margulis, Hagit Zer, Hagar Lis, Hanan Schoffman, Omer Murik, Ginga Shimakawa, Anja Krieger-Liszkay, Nir Keren

**Affiliations:** 1Department of Plant and Environmental Sciences, Edmond J. Safra Campus, The Alexander Silberman Institute of Life Sciences, Hebrew University of Jerusalem, Givat Ram, 9190402 Jerusalem, Israel; ketty.margulis@mail.huji.ac.il (K.M.); hagit.zer@mail.huji.ac.il (H.Z.); hagarlis@gmail.com (H.L.); hanan.schoffman@mail.huji.ac.il (H.S.); 2Medical Genetics Institute, Shaare Zedek Medical Center, 9103102 Jerusalem, Israel; omer.murik@mail.huji.ac.il; 3Institute for Integrative Biology of the Cell (I2BC), Commissariat à l’Energie Atomique et aux Energies Alternatives (CEA) Saclay, Centre National de la Recherche Scientifique (CNRS), Université Paris-Saclay, 91198 Gif-sur-Yvette CEDEX, France; ginshimakawa@gmail.com (G.S.); Anja.LISZKAY@i2bc.paris-saclay.fr (A.K.-L.)

**Keywords:** cyanobacteria, electron transport, photosynthesis, carbon metabolism, redox

## Abstract

Pgr5 proteins play a major direct role in cyclic electron flow paths in plants and eukaryotic phytoplankton. The genomes of many cyanobacterial species code for Pgr5-like proteins but their function is still uncertain. Here, we present evidence that supports a link between the Synechocystis sp. PCC6803 Pgr5-like protein and the regulation of intracellular redox balance. The knockout strain, *pgr5*KO, did not display substantial phenotypic response under our experimental conditions, confirming results obtained in earlier studies. However, the overexpression strain, *pgr5*OE, accumulated 2.5-fold more chlorophyll than the wild type and displayed increased content of photosystems matching the chlorophyll increase. As a result, electron transfer rates through the photosynthetic apparatus of *pgr5*OE increased, as did the amount of energy stored as glycogen. While, under photoautotrophic conditions, this metabolic difference had only minor effects, under mixotrophic conditions, *pgr5*OE cultures collapsed. Interestingly, this specific phenotype of *pgr5*OE mutants displayed a tendency for reverting, and cultures which previously collapsed in the presence of glucose were now able to survive. DNA sequencing of a *pgr5*OE strain revealed a second site suppression mutation in *slr1916*, a putative esterase associated with redox regulation. The phenotype of the *slr1916* knockout is very similar to that of the strain reported here and to that of the pmgA regulator knockout. These data demonstrate that, in Synechocystis 6803, there is strong selection against overexpression of the Pgr5-like protein. The pseudoreversion event in a gene involved in redox regulation suggests a connection of the Pgr5-like protein to this network.

## 1. Introduction

The major pathway of energy flow through the photosynthetic apparatus begins with light absorption by antenna pigment–protein complexes, leading to charge separation in photosystem reaction centers, and extends to the electron and proton transport process, finally generating NADPH and ATP [1]. This linear pathway is tightly regulated to avoid over-reduction of intermediate electron carriers and to balance ATP/NADPH ratios. Regulatory mechanisms identified at the level of the antenna systems include non-photochemical quenching processes [2,3,4,5]. Ion transporters regulate the proton gradient across the thylakoid membrane [6].

In addition to these, a number of cyclic, pseudo-cyclic and alternative electron flow pathways regulate electron transport processes [7,8]. Cyclic electron transport pathways are defined as reactions that involve only photosystem I (PSI) photochemistry. They support the production of a ΔpH without producing NADPH. Two such major pathways which funnel electrons from reduced ferredoxins (Fd) back to the plastoquinone (PQ) pool [9] have been identified in chloroplasts: one through the action of the chloroplast NDH (NADH dehydrogenase-like) complex and the other via PGR5/PGRL1 systems.

Chloroplasts contain a homologue of the mitochondrial NADH dehydrogenase complex. The chloroplast complex (the NADH dehydrogenase-like complex) was shown to be involved in cyclic electron flow. Current data suggest that, unlike its mitochondrial counterpart, the chloroplast NDH complex is able to interact directly with Fd [10,11,12]. In angiosperms, this route is considered to be minor. The major cyclic route in angiosperms is suggested to take place via PGR5/PGRL1 proteins [13]. The PGR5 mutant (proton gradient regulator) was identified in a screen for chlorophyll fluorescence phenotypes in Arabidopsis [14]. The picture was completed with the identification of its membrane embedded counterpart PGRL1 [13]. A putative ferredoxin:plastoquinone reductase (FQR) was also suggested as a path for PQ reduction in chloroplasts [15]. Together, they were mapped to the antimycin A sensitive electron transport pathway that was originally discovered by Arnon and coworkers in the 1950s [16].

The phenotypes arising from disruptions of these genes are varied and depend on the species of photosynthetic organisms and their growth conditions, but they are all related to sensitivity under excess or fluctuating light conditions [17,18]. The study of the function of PGR5/PGRL1 is still ongoing and hypotheses for their function range from direct and substantial involvement in cyclic electron transfer to regulatory functions [19,20,21].

The photosynthetic pathway in chloroplasts is relatively simple as compared to that of cyanobacteria, where respiratory and photosynthetic electron transport pathways occur in the same compartment and intersect at the PQ pool [22,23]. Apart from photosystem II (PSII), the PQ pool can be reduced by respiratory succinate dehydrogenase (SDH) and NAD(P)H dehydrogenases (both NDH-1 and NDH-2) [24]. Furthermore, several cyanobacterial species, including *Synechocystis* sp. PCC6803, have sulfide quinone reductase (SQR) genes [25,26]. The PQ pool can be oxidized via the linear cytochrome *b*_6_*f*-plastocyanin route, which, in cyanobacteria, is shared between photosynthetic and respiratory electron transport chains. It can also be oxidized by respiratory terminal oxidases such as quinol oxidase (Cyd), alternate respiratory terminal oxidases (ARTO) or plastoquinol terminal oxidase (PTOX) that were shown to be able to accept electrons directly from PQ [27,28,29,30,31]. This maze of pathways opens up a range of possible routes for cyclic electron flow intertwined amongst the photosynthetic and respiratory processes. To simplify the discussion in this manuscript, we will refer to the electron transport pathway starting at PSII and ending at NADPH as linear and to other pathways connecting to the PQ pool as alternatives to the linear path.

The discovery of the plant PGR5/PGRL1 pathway sparked interest in homologous proteins in cyanobacteria. The genomes of some cyanobacteria code for proteins that show a certain degree of similarity to plant PGR5 (*ssr2016* in *Synechocystis sp*. PCC6803; [32]). A number of studies indicated that the pgr5-like gene is expressed under oxidative stress conditions [33,34]. It was also suggested that it is part of the regulon controlled by Hik33 and PerR response regulators [35]. However, deletion mutants in the Pgr5-like protein resulted in very minor phenotypic responses, as compared to the M55 mutant of the NDH-1 complex [36,37]. The most detailed analysis was performed by Yeremenko and coworkers [32], who were able to detect an antimycin A dependent effect on electron transport in a Δ*pgr5*/M55 double knockout strain. In addition, the double mutant exhibited light sensitivity. The identity of the PGRL1 counterpart of the cyanobacterial Pgr5-like protein is still unresolved. A recent study suggested that, in Synechocystis 6803, ORF Sll1217 may play this role [38].

In our studies, we identified a locus controlling the expression of the Pgr5-like gene. Disruption of a previously uncharacterized putative two-component system gene, *slr1658*, reduced the ability to recover from iron limitation [39]. Transcriptomic analysis of *slr1658* mutants placed Pgr5 as the topmost overexpressed protein. Interestingly, Pgr5 overexpression in these mutants was constitutive regardless of growth or external stress conditions. Here, we describe a study of both knockout and overexpression strains of the Pgr5 homologue in *Synechocystis* 6803 that provides insight into its functional importance in redox regulation.

## 2. Materials and Methods

### 2.1. Growth Conditions

Stock cultures of the glucose tolerant *Synechocystis* sp. strain PCC 6803 [40] (wild type (WT)) and mutants were grown in YBG11 (an EDTA (ethylenediaminetetraacetic acid) amended BG11 medium [41]), containing 6 μM iron. Stock cultures of *pgr5*KO and *pgr5OE* strains were supplemented with 50 μg/mL kanamycin. In several experiments, glucose was added to a final concentration of 5 mM, as indicated. Cultures were grown at 30 °C with constant shaking. Light intensity was set at 40 μmol photons m^−2^ s^−1^.

### 2.2. Strain Construction

Strains in this study were generated using the restriction–ligation method with Takara’s DNA Ligation Kit (Cat.# 6023) [42]. Restriction enzymes used in this work were SacI, SalI, SpeI and SacII by NEB for *pgr*5KO and NdeI and HpaI by NEB for *pgr5*OE. Kanamycin resistance caste was added for selection in both mutant strains. A map of the insertion sites is shown in Figure 1, and primer sequences are listed in Table 1. Transformation was performed as described in [43] for both strains. The vectors used for the construction were pGEM Teasy (Promega) for *pgr*5KO and pTKP2031v vector [44] for *pgr5*OE. The pTKP2031v vector carrying both upstream and coding region of the *slr2031* gene (Figure 1). The *slr2031* gene is not expressed and is often used as an insertion site [45]. The construct includes the strong constitutive promotor of *psbAII* upstream of the *slr2031* start codon, where a NdeI site was introduced. The *slr2031* gene contains a HpaI site, allowing the introduction of pgr5 gene (excised with NdeI and HpaI) within the open reading frame *slr2031*, as described by Satoh and coworkers [44]. It has been shown that no significant differences in chlorophyll content were observed in *Synechocystis sp*. strain PCC 6803 with *psbAII* promotor and a kanamycin resistance cascade compared to the wild type [44]. The *pgr5*OE vector was sequenced prior to introducing it to the genome in order to ensure its integrity (not shown).

### 2.3. Spectroscopy and Microscopy

Cell density was measured as OD_730_ [46] using a Carry 300Bio spectrophotometer (Varian, CA, USA). Additional measurements of cell density and size were performed by direct cell counting using a hemocytometer. PSI activity was measured as P700 photo-oxidation using a JTS-10 spectrophotometer [46], using 10 μM 3-(3,4-dichlorophenyl)-1,1- dimethylurea (DCMU) to block PSII electron transport and 10 μM 2,5-dibromo-3-methyl-6-isopropylbenzoquinone (DBMIB) to block cytochrome b6f electron transport. Photochemical efficiency (Fv/Fm) of PSII was measured using the Satlantic FIRe (Fluorescence Induction and Relaxation) System [47]. Chlorophyll fluorescence spectra at 77K was measured using a Quantamaster 8075 Spectrofluorometer (HORIBA Jobin Yvon PTI, NJ, USA) and NADPH oxidoreduction using the NADPH/9-AA module of a DUAL-PAM (Walz, Effeltrich, Germany).

### 2.4. Chlorophyll Extraction

Samples were centrifuged at 16,000× *g* for 10 min; 100% methanol was added to the pellet, and samples were incubated in the dark for 30 min and then spun down. Absorbance was measured using a Carry 300Bio spectrophotometer (Varian, CA, USA) at 665 nm. Chlorophyll concentrations were calculated according to Porra et al. [48].

### 2.5. Biochemical Assays

Protein quantification was conducted according to the Bradford method [49]. Samples for determining glycogen content were adjusted to OD_730_ 0.1 and then were collected after 3 days of growth with or without glucose. Cultures were washed two times with YBGll media before breakage. Glycogen concentrations were determined as described in [50] using the Glucose (GO) Assay Kit (Sigma Aldrich) (Cat. # G3660).

### 2.6. DNA Extraction, Library Preparation and Sequencing

DNA was extracted as described before [39]. Sequencing libraries were prepared using Celero DNA-seq kit (Tecan) and then sequenced on an Illumina HiSeq × 300 (2 × 150), generating 10 million paired-end reads per sample, giving an estimated average coverage of × 400.

The mutation was then verified by PCR (primers in Table 1), followed by Sanger sequencing.

### 2.7. Bioinformatics Analysis

Raw sequencing reads were filtered and adaptors trimmed using Trimmomatic with default parameters [51]. Quality of filtered reads was assessed using fastQC (http://www.bioinformatics.babraham.ac.uk/projects/fastqc/) [52]. The filtered reads were mapped to the reference genome (accession CP012832.1) using bwa [53]. Variations from the reference sequence were calculated using VarScan (Version 2.3.9) [54]. The effect of the called variants on the amino acid sequences was evaluated with bcftools [55].

### 2.8. RNA Extraction

Cultures of WT and *pgr5*OE were collected from the logarithmic growth phase and extracted as described before [39]. To avoid DNA contamination, RNA was treated with DNase using “TURBO DNA-free kit” ThermoFisher (Cat. # AM1907) and converted to cDNA using “RevertAID first strand cDNA synthesis kit“ ThermoFisher (Cat. # K1622). RT was then conducted using Hy-taq polymerase by Hylabs (Cat. # EZ1012) (primers used for RT in primer list—Table 1).

## 3. Results

Following up on the results from our previous work [39], we studied here the function of the *Synechocystis* 6803 PGR5-like protein. We constructed two strains: a deletion strain, *pgr5*KO, and an overexpression strain, *pgr5*OE (Figure 1A–D). The *pgr*5OE strain constitutively overexpressed the *pgr5* transcript (Figure 1E) at levels similar to those previously observed in the Δ*slr1658* strain [39]. Initially, we measured growth during the transition into and out of iron limitation. In these experiments, both mutants exhibited growth rates similar to those of the wild type (not shown). This indicated that the growth phenotype observed in the Δ*slr1658* cannot be recreated by overexpression of *pgr5* alone.

However, while the mechanism(s) responsible for the Δ*slr1658* phenotype remain to be identified, we did observe significant changes in the function of the photosynthetic apparatus in the *pgr5*OE strain. The cellular chlorophyll content was 2.5 times higher on average in the *pgr5*OE strain as compared to the wild type (Figure 2A). At the same time, the content of active PSI units increased. The maximal P700 oxidation value, ΔAmax [46], exhibited a similar 2.5 ratio to wild type values (Figure 2B). The fraction of electrons reducing PSI that arrive from alternative, non-PSII sources remained similar in standard YBG11 media (Figure 2C). PSII parameters, measured by fluorescence induction techniques, were also unchanged in both *pgr5*KO and *pgr5*OE strains as compared to wild type (Figure 2D, Appendix A).

Based on the role of PGR5 in plant systems and the observed effect on PSI in our experiments, we suspected a change in cellular redox regulation in *pgr5*OE mutants. To challenge the electron transport network, we measured the same parameters in mixotrophic cultures grown for three days in YBGII + 5 mM glucose. In the wild type, both chlorophyll and active PSI content increased under these conditions. *pgr5*KO strain values remained indistinguishable from wild type values. The *pgr5*OE strain retained 2.5 higher chlorophyll and active PSI parameters (Figure 2E,F). PSII fluorescence parameters were slightly lower than wild type and *pgr5*KO strains (Figure 2H). While, for these parameters, the effect of *pgr5*OE was not modified by the addition of glucose, the fraction of electrons reducing PSI that arrive from alternative, non-PSII sources increased dramatically in this strain and reached values of close to 40% of the electrons passing through PSI (Figure 2G).

We also examined the strains under high light conditions: WT and *pgr*5OE were very similar in their responses (Appendix A).

To show whether the ratio between PSI and PSII was altered in *pgr*5OE compared to wild type, we performed Western blot analysis, using antibodies directed against PsaA and PsbA as representatives of PSI and PSII. These blots indicated that, qualitatively, the levels of the two proteins changed in accordance with changes in the cellular chlorophyll content, indicating no alteration in PSI/PSII ratio in *pgr*5OE and wild type (Figure 3A,B). This observation was further supported by 77K chlorophyll fluorescence emission spectroscopy indicating constant PSI/PSII fluorescence intensity ratios regardless of the treatment (Figure 3C). In the absence of glucose, no significant change was observed between the curves. In the *pgr5*OE + Glu trace, there is a 2 nm blue-shift in the position of the PSI peak and an additional fluorescence band at ~655 nm not observed in the wild type. The PSI shift is consistent with partially assembled photosystems [56]. The 655 nm band excitation spectra were measured and a peak at 620 nm, corresponding to phycocyanin (PC) absorption, was observed (not shown). While the absorption cross-section of PC at 430 nm is low, the fluorescence intensity of uncoupled phycobilisome is high. Both changes are consistent with a deteriorating state of *pgr5*OE cells on glucose, indicating breakdown of photosystems and uncoupling of antenna complexes.

Downstream of PSI, we detected effects on the redox state of the NAD(P)H pool in the absence and in the presence of glucose (Figure 4A). In dark adapted glucose free (–Glu) cultures, the NAD(P)H pool was in a more oxidized state in wild type than in *pgr5*OE, as seen by the large increase in the fluorescence level upon the onset of light. In the presence of glucose, the NAD(P)H pool was almost fully reduced in the dark in the wild type, while *pgr5*OE exhibited a more oxidized pool (Figure 4A). In combination, these results suggest significant electron flux through the photosynthetic apparatus, from both linear and alternative sources in *pgr5*OE cells. A major sink for excess energy sources in cyanobacterial cells is glycogen [57,58,59] and, indeed, *pgr5*OE contained more glycogen. This effect was amplified considerably by the addition of glucose to the media (Figure 4B).

The *pgr5*KO strain growth was identical to that of the wild type in glucose supplemented media (Figure 5). However, in the presence of glucose, *pgr5*OE cultures collapsed (Figure 5). This phenotype was observed both by optical density and microscopy cell counts. However, a turn of events occurred when we repeated the growth experiments—*pgr5*OE cultures stopped collapsing when exposed to 5 mM glucose (Appendix A). Interestingly, other aspects of the phenotype were retained: chlorophyll content and PSI activity per cell remained higher in the *pgr5*OE mutant (3 × 10^−11^ ± 4 × 10^−12^ and 1.8 × 10^−5^ ± 2 × 10^−6^ respectively). Resequencing of the *pgr5*OE strain revealed that it had adopted a second site suppression mutation. The mutation is a single codon substitution resulting in a phenylalanine to a serine mutation in *slr1916* (Figure 6), a protein that was previously identified as part of the redox regulatory pathway, with a similar phenotype to that of the *pmgA* mutant [60]. Under high light conditions, it is glucose sensitive and has high PSI and chlorophyll content [60]. Going back to our original glycerol *pgr5*OE stocks resulted in the same outcome—initial glucose sensitivity that was lost over time.

## 4. Discussion

Our study of the *Synechocystis* 6803 Pgr5-like protein resulted in a number of observations. (a) Its deletion did not result in any major observable effect. These results are in line with previously published data [32]. (b) Overexpression resulted in glucose sensitivity but this phenotype was not stable and reverted on two occasions. Since we observed recurring events of loss of glucose sensitivity, we suggest that there is a strong selection against the overexpression of the Pgr5-like protein. The polyploidy of the *Synechocystis* 6803 genome allows these events to progress slowly to the point of complete loss of the glucose sensitivity phenotype. (c) Genomic analysis of one of the reversion events led to the identification of a second site suppressor mutation in *slr1916,* coding for a protein involved in redox regulation. The similarity in the phenotype of the *pgr5*OE strain (harboring the *slr1916* point mutation; Figure 6) and the *slr1916* knockout strain [60] raises the distinct possibility that the additional aspect of the phenotype is a result of the point mutation rather than the over expression.

*slr1916* is a very interesting gene with respect to redox regulation. It was annotated as an esterase, identified in a transposon library screen [60]. Disruption of *slr1916* resulted in an altered chlorophyll fluorescence kinetic profile, high chlorophyll and PSI content under high light and glucose sensitivity [60], similar to that observed in *pmgA* [61,62,63]. It was shown that *slr1916* is essential for growth under photomixotrophic conditions [64]. *slr1916* is strongly induced under different types of environmental stress (acid and heat shock stresses) [65,66].

To better understand these observations, we used Robetta to predict the structure of WT Slr1916 protein [67]. In all top predictions, F242, the amino acid that corresponds to the point mutation discussed above, is located on the surface of the Slr1619 protein (details in Figure 6). According to mCSM, a sever that evaluates the potential effect of mutations on protein stability [68], a mutation of F242 to serine would significantly destabilize Slr1619′s structure. The phenotype observed on the *pgr5*OE background could, therefore, be the result of an inability of Slr1916 to interact with binding partners to perform its function in redox regulation.

Pseudoreversions are not unique to this study as they were observed in numerous mutants involved in photosynthetic redox regulation as well as mutants downstream of the photosynthetic pathway [69,70]. For example, growth of the knockout strain of the *pgmA* redox regulator on glucose results in numerous pseudorevertant colonies, mostly mutated in NDH-1 complex components [71]. This is not surprising, as the tendency of the original glucose-sensitive *Synechocystis* 6803 strain to revert when exposed to glucose is what made it an appealing model system in the early days of cyanobacterial genetic research [72]. While this may be frustrating, we nevertheless argue that it is not arbitrary. The tendency of mutant strains in *pmgA* [71], *Slr1658* [39], *pgr5*OE and of other redox regulation related mutants to succumb to pseudoreversions testifies to their pivotal role. This tendency must be taken into account when considering reports on their physiological importance in this organism.

Failing to regulate energy flow can have detrimental effects, leading to cell death. However, the main cause for this cytotoxicity in glucose sensitive mutants is still being elucidated [73]. The discovery of a relation of Pgr5 to these processes adds another connection to this regulatory network controlling energy flow in cyanobacterial cells. This study also provides another example of the risk involved in the mutational analysis of major genes coding for proteins involved in redox regulation.

## Figures and Tables

**Figure 1 life-10-00174-f001:**
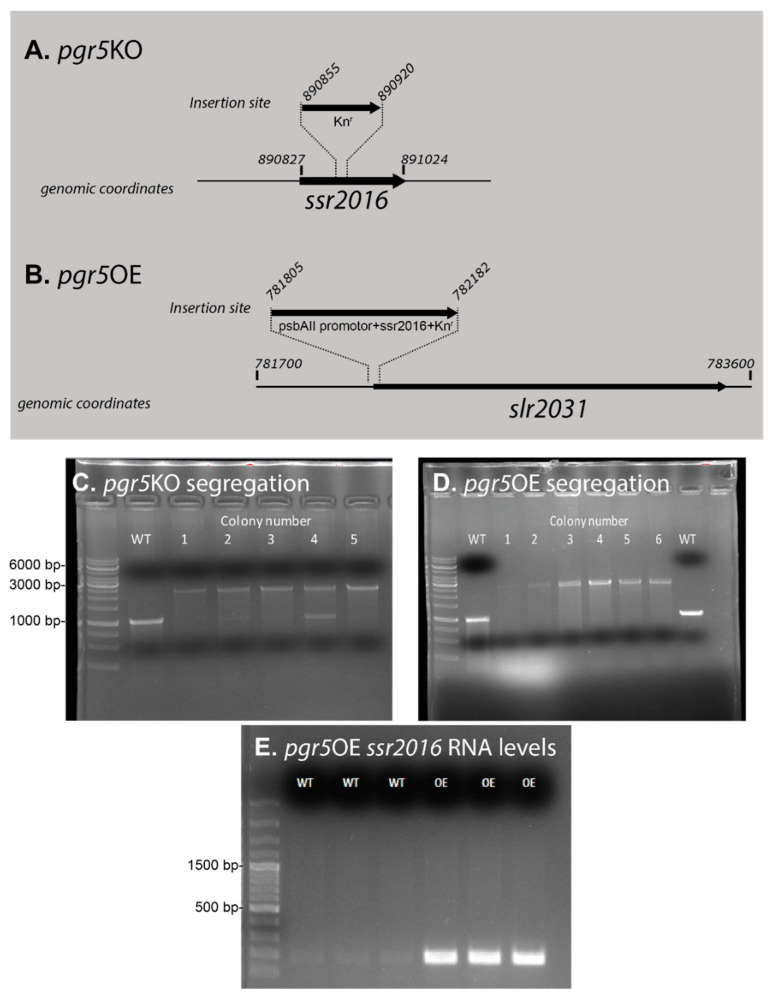
Genomic structure of mutants in the *ssr2016* gene. Panels (**A**) and (**B**) describe the genomic coordinates and insertion sites. The *slr2031* gene is not functional due to a deletion in its coding region [44]. The *pgr5*OE vector was sequenced prior to introducing it to the genome in order to ensure its integrity (not shown). Segregation of *pgr5*KO (wild type—~1000bp, *pgr5*KO—~2000bp) (**C**) and *pgr5*OE (wild type—~1000bp, *pgr5*OE—~3000bp) (**D**) was verified by PCR performed on genomic DNA (primer list in Table 1); GeneRuler 1 kb DNA Ladder—ThermoFisher (Cat. # SM0311) was used. Colony #5 of the *pgr5*KO and colony #4 of the *pgr5*OE were used for further work. (**E**) Amplified RNA expression of ssr2016 (~200bp) in wild type and *pgr*5OE (three repeats) by RT-PCR using the gene specific primers *pgr*5OE RT-PCR shown in Table 1. GeneRuler 100 bp plus DNA Ladder—ThermoFisher was used (Cat. # SM0321). Total RNA was extracted from cultures grown for three days in YBG11.

**Figure 2 life-10-00174-f002:**
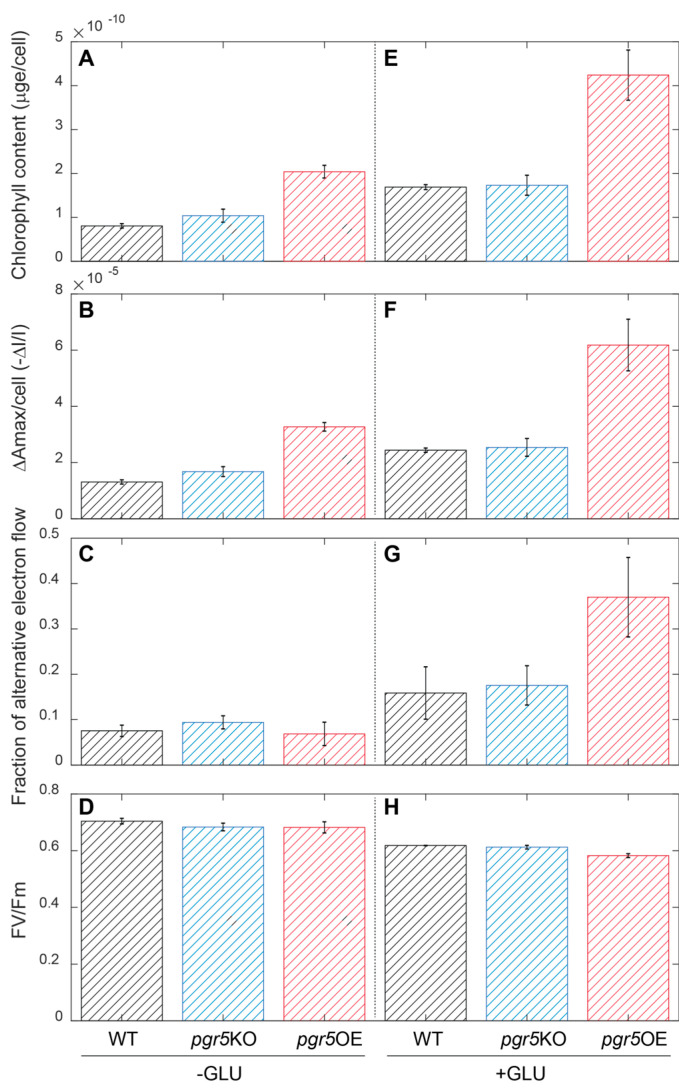
Photosynthetic parameters. Photosynthetic parameters were measured after 3 days of growth in YBG11 media (**A**–**D**) and in YBG11 media with 5 mM glucose (**E**–**H**). Chlorophyll extraction was performed and normalized to number of cells (**A**,**E**). The activity of PSI was measured as the maximal change in P700 absorbance—ΔAmax [46] (sample of raw data in Appendix A) normalized to number of cells (**B**,**F**). Alternative electron flow was calculated as the area trapped between the DCMU and DCMU and DBMIB measurements, normalized to the DCMU and DBMIB measurement (**C**,**G**). Fv/Fm of photosystem II was measured in 3 min dark adapted samples (**D**,**H**). Additional fluorescence parameters from the FIRe measurements are included in Appendix A. Error bars represent standard deviation with *n* = 3. Cell numbers for panels A-B, E-F: wild type 2.2 × 10^8^ ± 1.6 × 10^7^, *pgr5*KO 2.1 × 10^8^ ± 2.9 × 10^7^ and *pgr5*OE 6.9 × 10^7^ ± 8.8 × 10^6^. The average cell size of *pgr5*OE was larger than that of wild type (Appendix A).

**Figure 3 life-10-00174-f003:**
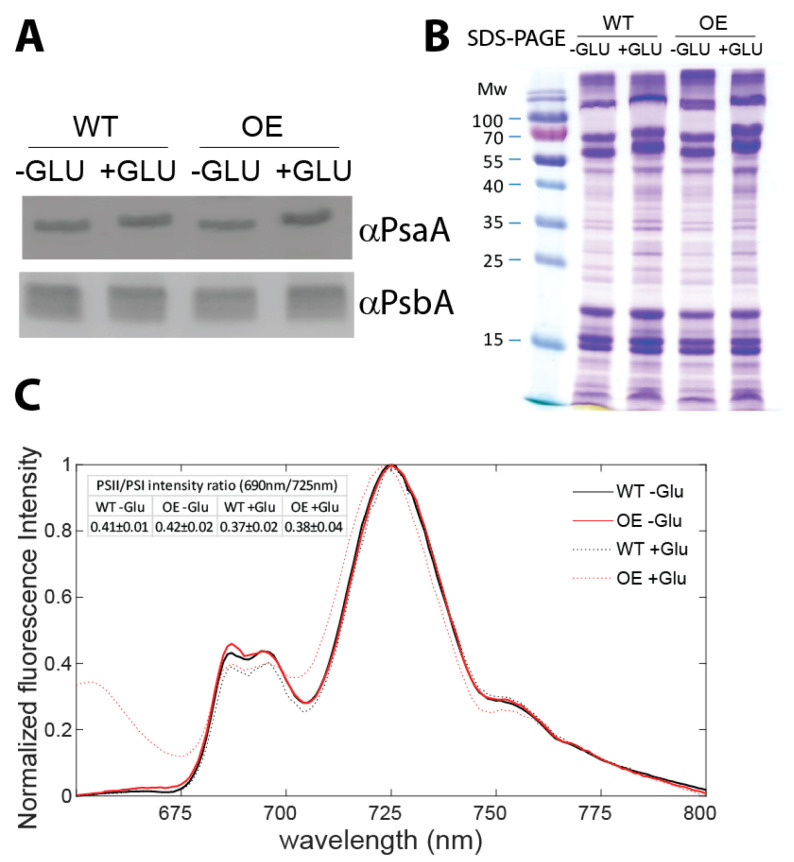
Photosystem ratios in the WT and in the *Pgr5*OE mutant. Western blot analysis using PsaA and PsbA antibodies (quantification of the blot is given in Appendix A) (**A**). SDS-PAGE analysis (**B**) of proteins in the WT and *pgr5*OE mutant. The gels were loaded on an equal chlorophyll basis (1.5 μg per lane). The experiment was repeated three times with comparable results (Appendix A). (**C**) 77 K chlorophyll emission fluorescence spectra. WT and *pgr5*OE cultures were grown on YBG11 medium with 5mM glucose for 3 days. Excitation at 430 ± 5 nm, mainly at the Soret band of chlorophyll but also exciting the far blue tail of phycocyanin absorption. Graphs are an average of three independent repeats. Statistical values for the ratio of PSII to PSI fluorescence intensity are presented in the inserted table.

**Figure 4 life-10-00174-f004:**
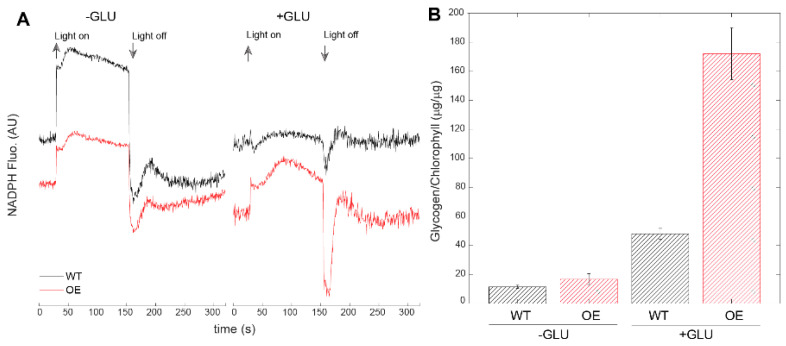
Energy flow downstream of PSI. (**A**) NAD(P)H fluorescence measurements. Black: control; red: *pgr5*OE. NAD(P)H fluorescence was measured in the dark and during exposure to actinic light (200 μmol photons m^−2^ s^−1^) of cultures with equal 4–5 µg chl ml^−1^ concentrations. Arrows indicate the illumination period. (**B**) Glycogen content measured in cells grown with and without glucose (5 mM) after 3 days of growth. Cellular glycogen concentrations were normalized to the chlorophyll concentration.

**Figure 5 life-10-00174-f005:**
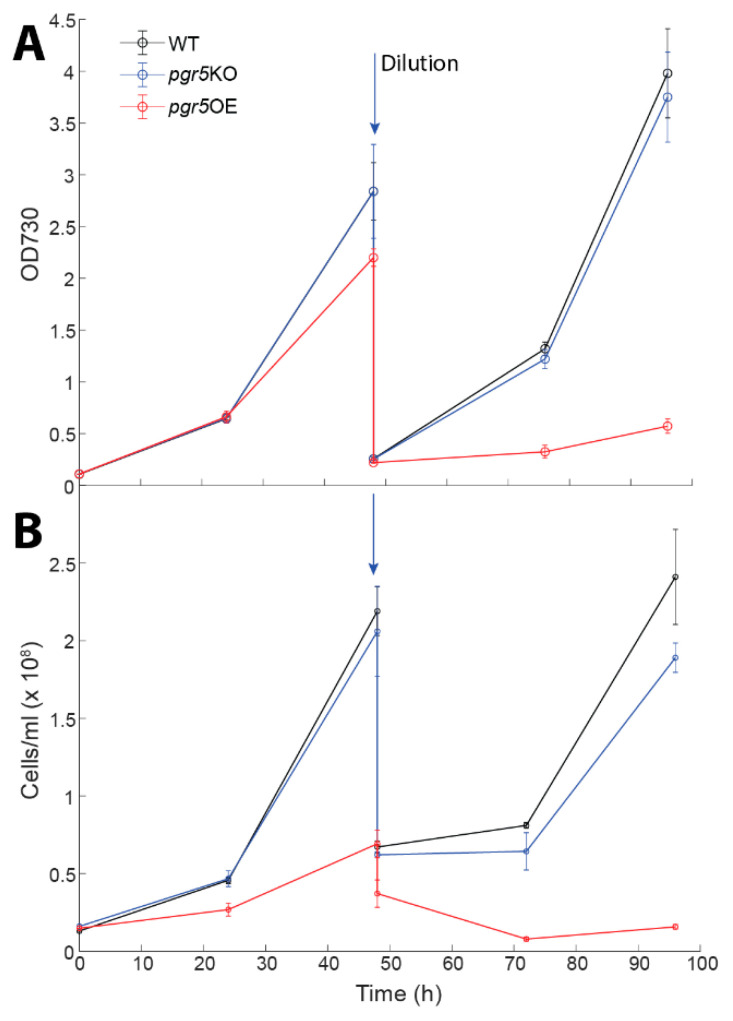
Effects of glucose on biomass accumulation. Cultures were grown on YBG11 medium with 5 mM glucose for five days. Biomass was monitored as optical density at 730 nm (**A**) and cells/mL (**B**). Error bars represent standard deviation derived from three repeats. Cultures were diluted to ensure that they remained in logarithmic phase and to ensure that the cultures did not exhaust media nutrients. The arrow marks the point at which the cultures were diluted.

**Figure 6 life-10-00174-f006:**
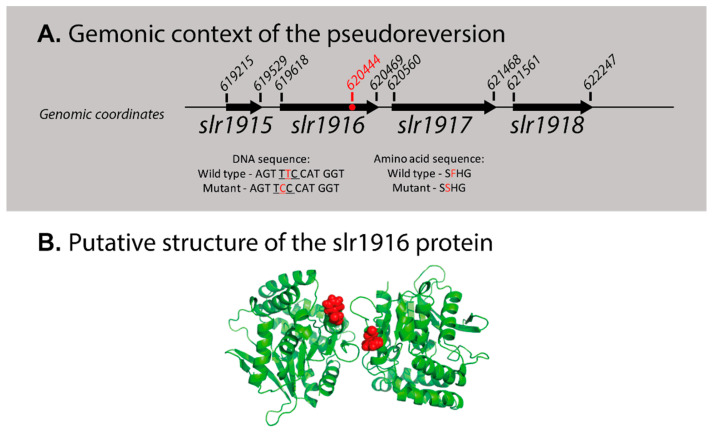
The genomic context and the putative structure of *slr1916*. (**A**) Genomic map—the start and end points of each gene are marked by their genomic coordinates. The *slr1916* missense mutation position is marked by a red dot. (**B**) The top scoring model from the Robetta analysis suggests that Slr1619 is a dimer. The mutated amino acid in the *pgr5*OE strain is shown in red.

**Table 1 life-10-00174-t001:** Primers.

Mutant Name	Genomic Location	Primer	
*pgr5*KO	890467–890486	F	5′-CACCATTGGCCTGGTATTGG-3′
	890848–890867	R	5′-TTGGTTCGTCAACAGTTAGG-3′
	890915–890938	F	5′-GCCAGACCATCACCAACTTTTGTA-3′
	891404–891424	R	5′-AAATGCCAGGTAACTAATTTG-3′
*pgr5*KO segregation check	890268–890287	F	5′-ACGTCACGTCCTTTGAGGTC-3′
	891290–891309	R	5′-GGATGACCAGGAAGCCAACC-3′
*pgr5*OE	890816–890835	F	5′-GAGTCACTGCCATGTTCGCC-3′
	891025–891049	R	5′-CTCTTCGTTTTCAATAATTCTTGCC-3′
*slr2030*–*slr2031* segregation check	781363–781383	F	5′-TGGGCACAACCATTTACCCTG-3′
	782328–782348	R	5′-AACTATGACCAACTGCGCCAG-3′
*pgr5*OE RT-PCR	890827–890852	F	5′-ATGTTCGCCCCCATCGTTATCTTGG-3′
	891289–891314	R	5′-GAGGGTTTTGCCGTTGGACTTAGCT-3′
*slr1916* mutation verification	619827–619847	F	5′-CCCGTTCAGAATATGACCTGG-3′
	620881–620902	R	5′-GCCGTACTTATTGGCAATTCC-3′

Primers used for this project.

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
