# Peer review of "Over Expression of the Cyanobacterial Pgr5-Homologue Leads to Pseudoreversion in a Gene Coding for a Putative Esterase in Synechocystis 6803"

_life, 2020, doi:10.3390/life10090174_

Round 1

Reviewer 1 Report

The manuscript Life -890614 Life from Margulis et al concerns the role of pgr5-like gene and the regulation of the intracellular redox energy in the model cyanobacterium Synechocystis sp. PCC 6803.

For that purpose they constructed a knockout mutant as well as a mutant over-expressing pgr5.

Using a combination of methods (RT PCR, spectroscopy for photosynthetic characteristics analysis, genome sequencing, glycogen dosage..) they analyzed the phenotype of these mutants under photoautotrophic growth conditions or in presence of glucose. The pgr5 overexpressing mutant is not growing well in the presence of glucose and during the culture some phenotypic revertants occurred. Therefore the authors sequenced the genome of this strain to identified the mutation allowing the culture to grow as well as the wid type. After sequencing the genome, they identify a mutation in an esterase encoding gene known to be essential for mixotrophic growth.

The experiments are well conducted and the results are very interesting. Their results shed light on the direction for further studies to improve the knowledge the regulation of energy flow in cyanobacteria.

However before to be published some revisions are necessary. To my opinion the manuscript has to be improved and to be written with more rigour.

Introduction

The introduction is well documented but may be more concise

Materials and methods

- Line 102: It is necessary to be clear and give more information concerning the “wild-type strain of Synechocystis sp. PCC 6803” used as the reference strain in this study). If I understand well, it is not the WT strain but a glucose tolerant strain with a deletion of 154 bp in the slr2031 gene.

Therefore the reference strain in this study is a mutant of the rbsU gene encoding a protein involved in the HL tolerance (Kamei et al 1998) and the regenerating growth after nitrogen or sulfur starvation and harbouring more pigments (carotenoids, phycocyanin, chlorophyll a) than the WT (Huckhauf et al. 2000).

- Lines 102-103: Why the authors used the YBG11?

   I suppose that the concentration of iron is 6 micromolar and not 6M.

-Line 107: the procedure to construct the deletion cassette or the overexpression cassette is not described in detail in Figure 1.

Did they construct the cassettes using Gibson assembly or using restriction-ligation?

Did they sequence the cassette before introduction into the genome to check the integrity of the sequence (no point mutation).

- Line 115: is it 10 M of DCMU and 10 M of DBMIB that were used?  

- Line 123: protein determination. Must be changed in Protein quantification

- Line 124: Samples collected after 3 days of growth: give more details (the initial OD or the cell number…)

-Line 125: some typographic errors as to be corrected: were washed twice with YBG11 medium before breakage.

Figure 1: there is a typographic error in the panel B. The coordinate at the beginning of slr2031 is the same that the coordinate of the beginning of insertion site (781653).

The DNA ladder or RNA ladder used are not indicated in the caption of the figure.

How were the RNA extracted, which enzyme was used for RT. This is not written in Figure 1, neither in the material and methods.

-Results :

- Lines 146-147: something is missing in front of slr1658 

- Figure 2 and line 147-148. The prgOE mutant harbors a chlorophyll content 2.5 times higher than the reference strain in photoautrophic conditions as well as in mixotrophic conditions.

There is an increase of chlorophll content in the presence of glucose in the WT strain  and the prgOE. Is this observed in other publications ? If not the authors should discuss this result.

-Line 151 WILD TYPE = Wild-type.

- Figure 3 and line 167 : Since the reference strain and the prgOE mutant grow at the same rate it will be logical to load the gel on protein quantity basis or cell count, and not chlorophyll, Since there is 2.5 more chlorophyll in the mutant.

This remark is also for the figure 4 panel B. The pgr5OE mutant is not growing well as shown in figure 5 but it is not clear why the authors dilute the culture after 2 days. They could have done a classical growth curve.

Because the authors observed a phenotypical reversion, they sequenced the genome of this revetant and find a mutation in the slr1916 gene. This is a very interesting experiment to identifiy interacting proteins or regulators.

They could also show a growth curve of the mutant that had acquiered the second mutation in the slr1916.

Author Response

This paper shows a study of PGR5-Like protein. This protein have been studied in detail in plants and there is scarce information in cyanobacteria. Authors shows that overexpresion of this protein resulted in glucose sensitivity and identified a supressor mutation in other ORF (slr1916).

The most interesting point is that the overexpression of PGR5 causes sensitivity to glucose involving the relationship of this protein with photosynthetic redox regulation.

I think it is necessary to check that this overexpressing mutant reverts in the presence of other stress conditions such as high light. 

The pgr5OE strain was tested for its response to high light. The results are included in the new supplementary file (Fig. S1). No significant difference was observed between pgr5OE and the wild type strain in this experiment. We also include now the data on an additional reversion event (Figs. S2). Our conclusion is that the expression of Pgr5 by itself is sufficient to trigger these events. The first paragraph of the discussion was rewritten to clarify this point.

Line 83 slr2016 is refered to Synechocytis – Done (line 88).

Line 92 ORF sll1217 – Done (line 96).

Line 108. It has been shown that overexpression of another protein under the psbA2 promoter does not produce an increase in the amount of chlorophyll?

It has been shown in Satoh et al. 2001 that the promoter-Knr construct does not result in an increase in the amount of chlorophyll. This is included in the methods section now (lines 120-122).

line 115 mM – Done (line 130).

line 116 mM – Done (line 131).

line 125 breakage – Done (line 145).

Figure 2- ug Chl/cell ....number of cells??

The data on the number of cells/ml is included in the figure legend.

Figure 3- How much chlorophyll is loaded per well?

1.5 mg per lane – added to figure legend.
Are there differences in some proteins in WT from EO? They could have been identified by Maldi-TOF.

This is a very interesting question, however, quantifying specific differences in photosystem protein abundance would require the production of strain with tags on different photosystem proteins as Liu et al. 2011. This is a major research effort that will constitute a project by itself.
Line 68 - The observed changes are not quantified.

We assume the referee is addressing line 168 instead of 68 as line 68 is in the introduction. We quantified the Western blot results and include data in supplemental table. S2.

Figure 4- Genomic not gemonic

We believe the referee found the mistake in figure 6 and not in figure 4. The mistake was corrected.

Line 150- wild type

The phrase “wild type” does not appear on line 150.

Line 151- WILD TYPE – Done (line 178).

Why does Overexpressing Mutant, despite having a higher amount of chlorophyll, show no change in Fv/Fm?

Fv/Fm is a relative proxy for the maximum quantum yield of PSII measured in dark adapted cells. There is no direct relation between this value and the amount of chlorophyll (Baker, N.R. and Oxborough, K., 2004).

Figure 4. With respect to NADPH it is not clear what differences are obtained in media without glucose between WT and OE.

In the absence of glucose, upon illumination the fluorescence increases drastically. This indicates that the NADP pool was largely oxidized in the dark. In the presence of glucose there was no increase upon illumination in the wild type and a small increase in the oePGR5 strain. (lines 212 – 213).

It is not clear whether the selective pressure is glucose or the over-expression of the PGR5 protein.

Our hypothesis is that the selective pressure is a result of the over expression itself. We rewrote the relevant text in the discussion (lines 237 – 239),

to make sure this point is clear (see our first comment above).  

Reviewer 2 Report

This paper shows a study of PGR5-Like protein. This protein have been studied in detail in plants and there is scarce information in cyanobacteria. Authors shows that overexpresion of this protein resulted in glucose sensivity and identified a supressor mutation in other ORF (slr1916).

The most interesting point is that the overexpression of PGR5 causes sensitivity to glucose involving the relationship of this protein with photosynthetic redox regulation.

I think it is necessary to check that this overexpressing mutant reverts in the presence of other stress conditions such as high light. 

Line 83 slr2016 is refered to Synechocytis

Line 92 ORF sll1217

Line 108. It has been shown that overexpression of another protein under the psbA2 promoter does not produce an increase in the amount of chlorophyll?

line 115 mM

line 116 mM

line 125 breakage

Figure 2- ug Chl/cell ....number of cells??

Figure 3- How much chlorophyll is loaded per well?
Are there differences in some proteins in WT from EO? They could have been identified by Maldi-TOF.

Line 68- The observed changes are not quantified.

Figure 4- Genomic not gemonic

Line 150- wild type

Line 151- WILD TYPE

Why does Overexpressing Mutant, despite having a higher amount of chlorophyll, show no change in Fv/Fm?

Figure 4. With respect to NADPH it is not clear what differences are obtained in  media without glucose between wt and OE.

It is not clear whether the selective pressure is glucose or the over-expression of the PGR5 protein.

Author Response

Answer to reviewer 2

The present work describes in brief the influence of both the knockout and over-expression of the slr1916 gene on the photosynthetic parameters of Synechocystis sp. strain PCC 6803. The authors founded the reverse of the known suppression effect of glucose in case of pgr5OE mutant (over-expression of the slr1916 gene) and suggested that it could be due to of inability of the slr1916 to interact with binding partners to perform its function in redox regulation.

We would like to point out that we did not over express slr1916. Pgr5OE mutant overexpresses pgr5 (ssr2016).

Broad comments.

The Introduction is written in good manner and takes into account all needed aspects for the present manuscript. I have some remarks only about using abbreviations, which have descriptions not after them, but below in the text. Additional remarks are about the description of NPQ. Usually, NPQ includes qT (state transitionS), qE (PsbS and Zea in higher plants; Lhcsr3, probably Lhcsr1 and Zea in algae and cyanobacteria), and qI (photoinhibition). It’s not clear why authors divided state transitions from NPQ? I can recommended reading and use more recent articles, such as Allorent et al.,2013 (10.1105/tpc.112.108274), Vecci et al., 2020 (10.3390/plants9010067), etc.

Agreed, we deleted state transition as it is included as part of NPQ. The paper by Allorent et al. is work done on Chlamydomonas. Vecci et al. is a review of microalgae using LHC systems. We limited our reference list to cyanobacterial literature, specifically to those with a PC-APC phycobilisome systems.

The Results section is written in a very short manner; regardless it presents a lot of data. Moreover, the major part is concerning of photosynthetic parameters between the studied strains in different conditions. This raises the question about the title of the manuscript. If based only on the Results, the title probably is needed to be changed. Additionally, it will be interesting to see the data about the size of the cells of studied strains, because the increase in chlorophyll content and PSI amount in OE per cell could be a result of the larger cell size. In this line, I recommended to add in Fig.3 densitograms of blots with errors. The authors showed the comparison of Maximal quantum yield or Maximal photosynthetic capacity of PSII (as Fv/Fm), however why they don’t show the comparison of other main photosynthetic parameters of PSII such as the Effective quantum yield (Y(II)), NPQ, etc? This is a very easy technic, especially with using DUAL-PAM.  This can help to see the real absence of differences in PSII operations under your light conditions.

  • Cell size data is included in the new supplementary file (Figure S4).
  • We did not use a DUAL-PAM. Generally, PAM methodology provides complicated results with cyanobacteria (See for example (Campbell et al. 1998). Instead we’ve used the FIRe approach which gave us much better results. A table of additional parameters form FIRe measurements is included in supplemental table S1. These include: The values of sPSII - Effective absorption cross section of PSII in darkness, t1 PSII – Rate constant for the minimum turnover time of PSII photochemistry after single saturating flash and t2 PSII – Rate constant for the minimum turnover time of PSII photochemistry after a train of multiple saturating flashes. These values were found to be similar in the wild type, pgr5OE and pgr5

The Discussion is surprisingly short. It mainly described the behavior of the slr1916 gene. However, it stayed understandable for me the reason why PSI and chlorophyll content were increased, while the PSII content and its capacity were the same in OE.  Additionally, the hypothesis about redox regulation (the Title has this phrase) is written complicated and unclear. I recommended to think about a schematic representation of the hypothesis here such as it was made in Eremenko et al., (2005), which is cited. It’s written, that the mutation makes impossible the interaction of the slr1916 with a binding partner, which is resulted in the observed phenotype. Which partner do you propose? Why it’s not the same when the slr1916 is knock out (KO)?

As we have stated in our replay to the first comment we did not knockout or overexpress slr1916. We suggest that the point mutation in slr1916 is a pseudoreversion event caused by the over expression of pgr5 (ssr2016).

Slr1916 was knocked out by Ozaki et al. 2007. The phenotype of the slr1916 knockout strain was very similar to the one we observed in the strain containing the slr1916 pseudoreversion. Slr1916 is a putative esterase and its binding partners are still unknown. We rewrote this paragraph in the discussion to make sure it is absolutely clear (lines 234 - 245).

Specific comments.

Line 44.    Usually used - state transitionS

State transitions were deleted (line 47).

Line 55.    The abbreviation NDH has a description only in line 71.

Corrected (line 54).

Line 58.    Incorrect writing. In Allahverdiyeva: “…screen based on its inability to perform non-photochemical quenching (NPQ)”.

This text does not appear in line 58. We would appreciate it if the referee can point out the exact location of the incorrect writing.

Line 86/91.    Pgr5 or PGR5?

PGR5 in vascular plant terminology. Pgr5 and pgr5, to denote protein or DNA respectively, in bacterial terminology. We made sure the use is consistent throughout the text.

Line 103.   “6 M iron” – I think that it’s due to the incorrect conversion to pdf, especially because the same in lines 115-116, but pay your attention at this moment.

Corrected (line 108).

Line 117.    Fv/Fm is the maximal quantum yield or maximal photosynthetic captures of PSII, but not the effective one.

Fv/Fm is a way of measuring the quantum yield of PSII. It is an indirect measure of the quantum yield and is influenced by a verity of factors including the dark level of PQ pool reduction (Baker, N.R. and Oxborough, K., 2004). In cyanobacteria, where respiration and photosynthesis both feed electrons into this pool this is a major issue (Vermaas  W.F.J.,  2001). Therefore, we refer to Fv/Fm as the effective or apparent quantum yield to express this uncertainty.

Line 123.    Is it mean the concentration or which kind of determination do you mean?

We corrected the phrase to “protein quantification”. (line 142)

Line 140.    Need to insert the reference to your previous work.

Done (line 166).

Line 146.    Which exactly phenotype do you mean? The description needs to be added.

We have changed “phenotype” to “growth phenotype” (line 172).

Line 149.   The chlorophyll content is not related to the photosynthetic activity, it’s more related to structural properties.

Yes, but PSI content and activity appearing one sentence afterwards are related to photosynthetic activity (line 177).

Line 153.   Only Fv/Fm was shown in the manuscript, which else parameters of PSII do you mean?

See our response above on Fv/FM values and the new Table S1.

Line 168.   I think it will be useful to insert densitograms of your blots in Fig 3.

Included, in the new Table S2.

Line 173. “The PSI shift is consistent with partially assembled systems”. A reference is needed.

A reference was added (line 204).

Line 174.   PC is plastocyanin?

No. PC is phycocyanin. Corrected. (line 206).

Line. 174. “The 655 nm band excitation spectra peaks at 620 nm (not shown), corresponding to PC absorption”- The sentence has to be reconstructed.

The sentence was restructured (lines 205 – 206).

Line 190. “Not shown”. Why? I think it will be better to show this line in Fig 5 using dot-line, for example.

Now shown in Figure S2.

Line 479.    The chlorophyll content is not a Photosynthetic activity. Change the title of the figure on Photosynthetic parameters, for example.

The title was changed to “photosynthetic parameters” (Fig. 2).

Line. 484.    Which one extraction did you use (acetone, methanol, ethanol)? Add this information to the Material and methods as well as cite the related article.

We’ve used methanol extraction – added to methods section (lines 137 – 140).

Line 489.    I think that more correct write like - ΔA705max.

Corrected (Fig. 3, line 518).

Line 492.   It’s unclear which exact area was used. Add the information about this method in the Material and methods section.

An example of the raw data for these measurements was added to clarify this point. New Supplemental Figure S3.

Line 497.   You need to write - Maximal photochemical efficiency or just a Fv/Fm value.

Corrected (Fig. 3, line 526).

Line 499.   It’s very strange and risky. Usually, the dark incubation of the samples before Fv/Fm measurement takes about 20-30 min (or even more) to balance all processes in thylakoid membranes and obtain the maximum value of Fm. You could detect not a maximum level of Fm in your case. For example, see Yamamoto and Shikanai, 2018 (10.1104/pp.18.01343). Also, I was surprised that you used the conference material as a reference for this method (45). It’s better to use articles or even manual for a PAM (for Dual-PAM).

This is absolutely true for plant systems. However, in cyanobacteria where respiration and photosynthesis interact with the same PQ pool even 20-30 minutes will not result in complete oxidation of the PQ pool. Therefore, it is standard in work on cyanobacteria to use much shorter dark adaptation periods.

Reference 45 was replaced with (Gorbunov, Kolber, and Falkowski 1999) for the FIRe methodology. We did not use PAM. 

Reviewer 3 Report

A brief summary.

The present work describes in brief the influence of both the knockout and over-expression of the slr1916 gene on the photosynthetic parameters of Synechocystis sp. strain PCC 6803. The authors founded the reverse of the known suppression effect of glucose in case of pgr5OE mutant (over-expression of the slr1916 gene) and suggested that it could be due to of inability of the slr1916 to interact with binding partners to perform its function in redox regulation.

Broad comments.

The Introduction is written in good manner and takes into account all needed aspects for the present manuscript. I have some remarks only about using abbreviations, which have descriptions not after them, but below in the text. Additional remarks are about the description of NPQ. Usually, NPQ includes qT (state transitionS), qE (PsbS and Zea in higher plants; Lhcsr3, probably Lhcsr1 and Zea in algae and cyanobacteria), and qI (photoinhibition). It’s not clear why authors divided state transitions from NPQ? I can recommended reading and use more recent articles, such as Allorent et al.,2013 (10.1105/tpc.112.108274), Vecci et al., 2020 (10.3390/plants9010067), etc.

The Results section is written in a very short manner, regardless it presents a lot of data. Moreover, the major part is concerning of photosynthetic parameters between the studied strains in different conditions. This raises the question about the title of the manuscript. If based only on the Results, the title probably is needed to be changed. Additionally, it will be interesting to see the data about the size of the cells of studied strains, because the increase in chlorophyll content and PSI amount in OE per cell could be a result of the larger cell size. In this line, I recommended to add in Fig.3 densitograms of blots with errors. The authors showed the comparison of Maximal quantum yield or Maximal photosynthetic capacity of PSII (as Fv/Fm), however why they don’t show the comparison of other main photosynthetic parameters of PSII such as the Effective quantum yield (Y(II)), NPQ, etc? This is a very easy technic, especially with using DUAL-PAM.  This can help to see the real absence of differences in PSII operations under your light conditions.

The Discussion is surprisingly short. It mainly described the behavior of the slr1916 gene. However, it stayed understandable for me the reason why PSI and chlorophyll content were increased, while the  PSII content and its capacity were the same in OE.  Additionally, the hypothesis about redox regulation (the Title has this phrase) is written complicated and unclear. I recommended to think about a schematic representation of the hypothesis here such as it was made in Eremenko et al., (2005), which is cited. It’s written, that the mutation makes impossible the interaction of the slr1916 with a binding partner, which is resulted in the observed phenotype. Which partner do you propose? Why it’s not the same when the slr1916 is knock out (KO)?

Specific comments.

Line 44.    Usually used - state transitionS

Line 55.    The abbreviation NDH has a description only in line 71.

Line 58.    Incorrect writing. In Allahverdiyeva: “…screen based on its inability to perform non-photochemical quenching (NPQ)”.

Line 86/91.    Pgr5 or PGR5?

Line 103.   “6 M iron” – I think that it’s due to the incorrect conversion to pdf, especially because the same in lines 115-116, but pay your attention at this moment.

Line 117.    Fv/Fm is the maximal quantum yield or maximal photosynthetic captures of PSII, but not the effective one.

Line 123.    Is it mean the concentration or which kind of determination do you mean?

Line 140.    Need to insert the reference to your previous work.

Line 146.    Which exactly phenotype do you mean? The description needs to be added.

Line 149.   The chlorophyll content is not related to the photosynthetic activity, it’s more related to structural properties.

Line 153.   Only Fv/Fm was shown in the manuscript, which else parameters of PSII do you mean?

Line 168.   I think it will be useful to insert densitograms of your blots in Fig 3.

Line 173. “The PSI shift is consistent with partially assembled systems”. A reference is needed.

Line 174.   PC is plastocyanin?

Line. 174.  “The 655 nm band excitation spectra peaks at 620 nm (not shown), corresponding to PC absorption”- The sentence has to be reconstructed.

Line 190. “Not shown”. Why? I think it will be better to show this line in Fig 5 using dot-line, for example.

Line 479.    The chlorophyll content is not a Photosynthetic activity. Change the title of the figure on Photosynthetic parameters, for example.

Line. 484.    Which one extraction did you use (acetone, methanol, ethanol)? Add this information to the Material and methods as well as cite the related article.

Line 489.    I think that more correct write like - ΔA705max.

Line 492.   It’s unclear which exact area was used. Add the information about this method in the Material and methods section.

Line 497.   You need to write - Maximal photochemical efficiency or just a Fv/Fm value.

Line 499.   It’s very strange and risky. Usually, the dark incubation of the samples before Fv/Fm measurement takes about 20-30 min (or even more) to balance all processes in thylakoid membranes and obtain the maximum value of Fm. You could detect not a maximum level of Fm in your case. For example, see Yamamoto and Shikanai, 2018 (10.1104/pp.18.01343). Also, I was surprised that you used the conference material as a reference for this method (45). It’s better to use articles or even manual for a PAM (for Dual-PAM).

Author Response

Answer to reviewer 3

The manuscript Life -890614 Life from Margulis et al concerns the role of pgr5-like gene and the regulation of the intracellular redox energy in the model cyanobacterium Synechocystis sp. PCC 6803.

For that purpose they constructed a knockout mutant as well as a mutant over-expressing pgr5.

Using a combination of methods (RT PCR, spectroscopy for photosynthetic characteristics analysis, genome sequencing, glycogen dosage..) they analyzed the phenotype of these mutants under photoautotrophic growth conditions or in presence of glucose. The pgr5 overexpressing mutant is not growing well in the presence of glucose and during the culture some phenotypic revertants occurred. Therefore the authors sequenced the genome of this strain to identified the mutation allowing the culture to grow as well as the wid type. After sequencing the genome, they identify a mutation in an esterase encoding gene known to be essential for mixotrophic growth.

The experiments are well conducted and the results are very interesting. Their results shed light on the direction for further studies to improve the knowledge the regulation of energy flow in cyanobacteria.

However before to be published some revisions are necessary. To my opinion the manuscript has to be improved and to be written with more rigour.

Introduction

The introduction is well documented but may be more concise

Materials and methods

- Line 102: It is necessary to be clear and give more information concerning the “wild-type strain of Synechocystis sp. PCC 6803” used as the reference strain in this study). If I understand well, it is not the WT strain but a glucose tolerant strain with a deletion of 154 bp in the slr2031 gene. Therefore the reference strain in this study is a mutant of the rbsU gene encoding a protein involved in the HL tolerance (Kamei et al 1998) and the regenerating growth after nitrogen or sulfur starvation and harbouring more pigments (carotenoids, phycocyanin, chlorophyll a) than the WT (Huckhauf et al. 2000).

The Synechocystis sp. PCC 6803 strain used in this study is glucose tolerant (Williams 1988) (line 107). The slr2031 site was used for the insertion of the pgr5OE construct (Figure 1). The methods were corrected (line 112 – 125).

- Lines 102-103: Why the authors used the YBG11?

YBG11 is an EDTA amended BG11 medium that we use to enable manipulation of trace metal concentrations (Methods line 108).

   I suppose that the concentration of iron is 6 micromolar and not 6M.

Corrected (line 108).

-Line 107: the procedure to construct the deletion cassette or the overexpression cassette is not described in detail in Figure 1.

This is included in the methods section now (lines 112 – 125).

Did they construct the cassettes using Gibson assembly or using restriction-ligation?

We used restriction- ligation method. methods section was updated (line 113).

Did they sequence the cassette before introduction into the genome to check the integrity of the sequence (no point mutation).

Yes, we did. The sequence was intact. The information was added to methods (line 124).

- Line 115: is it 10 M of DCMU and 10 M of DBMIB that were used? 

Corrected (line 130 – 131).

- Line 123: protein determination. Must be changed in Protein quantification

Corrected (line 142).

- Line 124: Samples collected after 3 days of growth: give more details (the initial OD or the cell number…)

Cells were adjusted to OD 730 0.1 on day zero and then collected after 3 days of growth (lines 143).

-Line 125: some typographic errors as to be corrected: were washed twice with YBG11 medium before breakage.

Corrected (line 145).

Figure 1: there is a typographic error in the panel B. The coordinate at the beginning of slr2031 is the same that the coordinate of the beginning of insertion site (781653).

We thank the reviewer for pointing this out. The insertion coordinates were corrected (Fig. 1).

The DNA ladder or RNA ladder used are not indicated in the caption of the figure.

Figure 1 and its legend of was updated to include information on the ladder (lines 498 - 506).

How were the RNA extracted, which enzyme was used for RT. This is not written in Figure 1, neither in the material and methods.

Methods section updated (lines 159 - 164).

-Results :

- Lines 146-147: something is missing in front of slr1658

Corrected (line 172 – 174).

- Figure 2 and line 147-148. The prgOE mutant harbors a chlorophyll content 2.5 times higher than the reference strain in photoautrophic conditions as well as in mixotrophic conditions. There is an increase of chlorophyll content in the presence of glucose in the WT strain and the pgrOE. Is this observed in other publications? If not the authors should discuss this result.

The increase in chlorophyll content in the presence of glucose in the media was observed at least once at (Elmorjani and Herdman 1987).  

-Line 151 WILD TYPE = Wild-type.

Corrected (line 178).

- Figure 3 and line 167: Since the reference strain and the pgrOE mutant grow at the same rate it will be logical to load the gel on protein quantity basis or cell count, and not chlorophyll, since there is 2.5 more chlorophyll in the mutant.

It is logical however, in our experience, it was practically more straightforward and accurate to load gels on a chlorophyll basis.

This remark is also for the figure 4 panel B. The pgr5OE mutant is not growing well as shown in figure 5 but it is not clear why the authors dilute the culture after 2 days. They could have done a classical growth curve.

Cultures were diluted to ensure they remain in logarithmic phase and to ensure the cultures do not exhaust media nutrients (Fig.5 legend was corrected to include this point).

Because the authors observed a phenotypical reversion, they sequenced the genome of this revetant and find a mutation in the slr1916 gene. This is a very interesting experiment to identify interacting proteins or regulators.

They could also show a growth curve of the mutant that had acquired the second mutation in the slr1916.

This is now shown in figure S2.

Round 2

Reviewer 3 Report

I thank the authors for taking into account the main part of the previous comments and improving the quality of the manuscript.

Nevertheless, I still have some minor comments and remarks about the manuscript.

First of all, it was clear that you didn’t overexpress the protein, but the overexpression of the protein pgr5 occurred in the mutant and had an influence on the photosynthetic parameters.  

Me: Line 58.    Incorrect writing. In Allahverdiyeva: “…screen based on its inability to perform non-photochemical quenching (NPQ)”.

Replay: This text does not appear in line 58. We would appreciate it if the referee can point out the exact location of the incorrect writing.

The new version of the manuscript lines 60-61: The PGR5 mutant (proton gradient regulator) was identified in a screen for fluorescence quenching phenotypes in Arabidopsis [14].

I think it is written incorrectly because if take the original figure 1 from Munekage et al., 2002, which work is 14 in your reference list, the pgr5 mutant showed “High chlorophyll fluorescence phenotype”, but not fluorescence quenching phenotypes. In Allahverdieva et al., 2014 (p. 4): “Originally, the pgr5 mutant was identified in a mutant screen based on its inability to perform non-photochemical quenching (NPQ) upon exposure to high light (Munekage et al., 2002)”. Thus, the fluorescence quenching is not a phenotype, but the ability. You have to probably correct the sentence to fully match the text you cite.

In the description of the chlorophyll extraction procedure, it’s important to indicate the exact concentration of the used methanol in % due to the dependence of the extinction coefficients on it, as well as to refer the articles described the method.

In Line 194 you should write PsbA instead of PsaB, because the PsaB is some protein of PSI. I think it’s just a mistake, moreover, it is showed correctly in Fig.3,A. You have to be more attentive.

It’s really surprising that statistical errors in the quantification of western blots are near 30-43% (Table S2) in 3 out of 6 cases of experiments, and one more experiment has an error of 21%. I believe that these data do not allow making any conclusions.

You didn’t change in the presented manuscript, despite your answer, the ΔAmax to ΔA705max in Fig.3 and its caption. However, I still think you should do it.

In supplementary material in Fig S2 it would be useful to describe that the symbol above the panels is the date. In addition, to draw the arrows, which are mentioned in the caption.

Author Response

Me: Line 58.    Incorrect writing. In Allahverdiyeva: “…screen based on its inability to perform non-photochemical quenching (NPQ)”.

Replay: This text does not appear in line 58. We would appreciate it if the referee can point out the exact location of the incorrect writing.

The new version of the manuscript lines 60-61: The PGR5 mutant (proton gradient regulator) was identified in a screen for fluorescence quenching phenotypes in Arabidopsis [14].

I think it is written incorrectly because if take the original figure 1 from Munekage et al., 2002, which work is 14 in your reference list, the pgr5 mutant showed “High chlorophyll fluorescence phenotype”, but not fluorescence quenching phenotypes. In Allahverdieva et al., 2014 (p. 4): “Originally, the pgr5 mutant was identified in a mutant screen based on its inability to perform non-photochemical quenching (NPQ) upon exposure to high light (Munekage et al., 2002)”. Thus, the fluorescence quenching is not a phenotype, but the ability. You have to probably correct the sentence to fully match the text you cite.

The sentence was corrected. We changed to “The PGR5 mutant (proton gradient regulator) was identified in a screen for chlorophyll fluorescence phenotypes in Arabidopsis”. (Lines 60-61). 

In the description of the chlorophyll extraction procedure, it’s important to indicate the exact concentration of the used methanol in % due to the dependence of the extinction coefficients on it, as well as to refer the articles described the method.

We used 100% methanol for the extraction. Methods wad updated and a reference to Porra et al. 1989 was added (lines 137-140). 

In Line 194 you should write PsbA instead of PsaB, because the PsaB is some protein of PSI. I think it’s just a mistake, moreover, it is showed correctly in Fig.3,A. You have to be more attentive.

Thank you, corrected. (line 195).

It’s really surprising that statistical errors in the quantification of western blots are near 30-43% (Table S2) in 3 out of 6 cases of experiments, and one more experiment has an error of 21%. I believe that these data do not allow making any conclusions.

The accuracy of data in western blots generates qualitative results. It is very hard, even in the best cases, to extract more accurate quantitative data from this technique. This is indicated in the text “These blots indicated that, qualitatively, the levels of the two proteins changed in accordance with changes in the cellular chlorophyll content” (lines 196-198).

You didn’t change in the presented manuscript, despite your answer, the ΔAmax to ΔA705max in Fig.3 and its caption. However, I still think you should do it.

The term “DAmax” was coined in a previous paper from our group (Salomon el al. Plant Physiology, 2011). We want to stay consistent with our previous papers.

In supplementary material in Fig S2 it would be useful to describe that the symbol above the panels is the date. In addition, to draw the arrows, which are mentioned in the caption.

Thank you again, we corrected the figure. (Fig S2).